# Epidemiology of Oral Cancer in Taiwan: A Population-Based Cancer Registry Study

**DOI:** 10.3390/cancers15072175

**Published:** 2023-04-06

**Authors:** Chao-Wei Chou, Chun-Ru Lin, Yi-Ting Chung, Chin-Sheng Tang

**Affiliations:** 1School of Public Health, College of Medicine, Fu Jen Catholic University, No. 510, Zhongzheng Road, Xinzhuang District, New Taipei City 242, Taiwan; 407500706@mail.fju.edu.tw (C.-W.C.); 407500653@mail.fju.edu.tw (Y.-T.C.); 2Department of Medical Education, Chang Gung Memorial Hospital, Linkou Branch, No. 5, Fuxing Street, Guishan District, Taoyuan City 333, Taiwan; chunru0725@cgmh.org.tw

**Keywords:** oral cancer, secular trends, age-standard incidence rate

## Abstract

**Simple Summary:**

Oral cancer (OC) is prevalent cancer worldwide, with varying incidence rates in different regions. To analyze the secular trend of the incidence of OC in Taiwan, this study used data from the national cancer registry database between 1980 and 2019. The study used the age-period-cohort model and average annual percentage change (AAPC) to examine the characteristic of incidence. The study also used Spearman’s correlation to determine the correlation between age-standard incidence rates (ASR) of OC and related risk factors. Results indicated that ASR of OC increased in men and women from 1980–1984 to 2015–2019. The age-period-cohort model revealed a peak incidence rate in the 1975 cohort in men, followed by a declining trend. The study found a correlation between the incidence of OC and changes in cigarette and alcohol consumption and the production of betel quid. Therefore, the study recommends avoiding these risk factors to prevent OC.

**Abstract:**

Oral cancer (OC) is one of the most common cancers worldwide, and its incidence has regional differences. In this study, the cancer registry database obtained from 1980 to 2019 was used to analyze the characteristic of incidence of OC by average annual percentage change (AAPC) and an age–period–cohort model. Spearman’s correlation was used to analyze the relationship between the age-standard incidence rates (ASR) of OC and related risk factors. Our results showed that the ASR of OC increased from 4.19 to 27.19 per 100,000 population with an AAPC of 5.1% (95% CI = 3.9–6.3, *p* value < 0.001) in men and from 1.16 to 2.8 per 100,000 population with an AAPC of 3.1% (95% CI = 2.6–3.6, *p* value < 0.001) in women between 1980–1984 and 2015–2019. The age–period–cohort model reported a trend of rising then declining for the rate ratio in men, with peaks occurring in the 1975 cohort, with a rate ratio of 6.80. The trend of incidence of oral cancer was related to changes in the consumption of cigarettes and alcohol and production of betel quid, with r values of 0.952, 0.979 and 0.963, respectively (all *p* values < 0.001). We strongly suggest avoiding these risk factors in order to prevent OC.

## 1. Introduction

Based on the Global Cancer Statistics 2020, oral cancer is one of the most common cancers worldwide. The global age-standardized rate (ASR) of oral cancer was 6.0 per 10,000 in males and 2.3 per 10,000 in females. However, the incidence of oral cancer has regional variation. Among the six continents, Asia had the highest incidence of oral cancer (65.8%), followed by Europe (17.3%) and North America (7.3%) [1]. According to the Human Development Index of the United Nations Development Program, the incidence of oral cancer is higher in countries with a better development index, while mortality is higher in less developed areas, which indicates social inequality [2]. Oral cancer has multiple risk factors, including smoking, other tobacco consumption, snuff dipping, alcohol, sunlight exposure, viruses, and so on [3]. The most common pathology of oral cancer is squamous cell carcinoma, and the major risk factors are tobacco and alcohol [4]. A previous cohort study with 177,271 adult men reported that the mortality hazard ratio (HR) was 12.52 for oral cancer in chewers of betel quid [5]. Surgical resection with or without postoperative radiation or chemoradiation therapy is the standard treatment for the management of oral cancer [4]. Based on the Global Cancer Statistics 2020, the age-standardized mortality of cancer of the lip and oral cavity was 2.8 per 100,000 in men and 1.0 per 100,000 in women worldwide [1,3,6]. In many countries, including the USA, population-based screening for oral cancer is not recommended due to insufficient evidence demonstrating its efficacy in reducing mortality [7]. Taiwan has been conducting a national population-based screening program for oral cancer since 2004. From 2004 to 2009, the screening rate was 55.1%, and mortality decreased by 26% in the screened group [8]. The epidemiology of oral cancer and the relationship between oral cancer and its risk factors may differ in different countries and at different times. For example, betel leaf and areca nut consumption are common social practices in South Asia, Southeast Asia, and Pacific Asia, as well as in emigrated communities in North America and Europe, and this has been identified as a risk factor for head and neck cancers [9]. Due to the serious threat of oral cancer to public health, this study aimed to offer the latest information regarding public health based on a population-based database to analyze the age–period–cohort (APC) effect of oral cancer.

## 2. Materials and Methods

### 2.1. Data Sources

This research was an observational study. The number of newly diagnosed oral cancer cases in each age group (0–89 years) from 1980 to 2019 was retrieved from the Taiwan Cancer Registry (TCR), a nationwide population-based cancer registry system maintained by the Health Promotion Administration, and all data in this study were obtained from the public website without personal information [10]. Oral cancer is defined as cancer developing in the oral cavity, oropharynx, and hypopharynx. We used the International Classification of Diseases for Oncology Field Trial Edition (ICD-O-FT) codes 140–149 before 2002, and the International Classification of Diseases for Oncology (ICD-O-3) codes C00-C14 after 2002 to identify oral cancer cases and excluded salivary gland malignancies (ICD-O-FT code: 142 and ICD-O-3 code: C07-C08) and nasopharyngeal carcinoma (ICD-O-FT code: 147 and ICD-O-3 code: C11). The TCR was developed in 1979 by the Ministry of Health and Welfare to establish a long-term, complete, and accurate cancer database. The percentage of death certificate only (DCO%) of all cancers decreased from 8.84% in 1998 to 0.71% in 2019, and the morphological verification percentage (MV%) in 2019 was 93.47% (men: 92.31%, women: 94.77%) [11].

Data on the consumption per person of tobacco and alcohol in this study were retrieved from the Taiwan Tobacco and Liquor Corporation survey [12] and the Ministry of Finance, Taiwan [13], respectively. Data on the production of betel nuts were obtained from the Council of Agriculture, Executive Yuan, Taiwan [14].

### 2.2. Statistical Analysis

The study analyzed the incidence rates of oral cancers over the period from 1980 to 2019, stratified by age groups ranging from 30–34 to 80–84 years of age and time periods ranging from 1980–1984 to 2015–2019. The birth cohorts were categorized into 18 groups, ranging from the oldest cohort (born in 1900–1904) to the youngest cohort (born in 1985–1989). The age group of 30–34 was chosen as the starting point for the graphs due to the low incidence of oral cancer among individuals under 30 years old. The age-adjusted incidence rates (ASR) were calculated using the 2000 World Standard Population [15]. To assess the secular trend in oral cancer incidence, the study utilized the National Cancer Institute’s online tool to calculate the average annual percentage change (AAPC) [16]. Additionally, the study investigated the period and cohort effects on oral cancer incidence through age–period–cohort modeling using the same web tool. This approach provided the rate ratios (RRs) that compared the incidence of oral cancers in different periods (period effects) and cohorts (cohorts) to the reference points [16]. The relationship between ASR of oral cancer and related risk factors was analyzed using Spearman’s correlation, and statistical analysis was conducted by Microsoft™ Excel™ 365 MSO 16.0.13528.203018 64bit (Microsoft Corporation, Redmond, Washington, WA, USA).

The research protocol was approved by the Institutional Review Board of Fu-Jen Catholic University (No. C110216).

## 3. Results

### 3.1. Trend of Incidence Rate of Oral Cancer

Figure 1 shows the trend of the ASR of oral cancer in men and women in Taiwan from 1980–1984 to 2015–2019. The ASR of men increased from 4.19 per 100,000 population in 1980 to a peak of 28.59 per 100,000 population in 2014 and was 27.19 per 100,000 population in 2019. The incidence of oral cancer in men tended to plateau after 2008, and its AAPC from 1980 to 2019 was 5.1% (95% CI = 3.9–6.3, *p* value < 0.001). The ASR of women increased from 1.16 per 100,000 population in 1980 to a peak of 3.15 per 100,000 population in 2017 and was 2.82 per 100,000 population in 2019, and its AAPC from 1980 to 2019 was 3.1% (95% CI = 2.6–3.6, *p* value < 0.001). The incidence of oral cancer in men displayed an obvious increase, while that in women exhibited a relatively gradual increase, with the incidence rate being higher in men than in women.

### 3.2. Age-Specific Incidence Rate by Different Periods

Figure 2a shows the age-specific rates of oral cancer in males per 100,000 men in Taiwan. The age-specific rate of oral cancer in males increased steadily between 1980–1984 and 2015–2019. The relative percent changes were 810%, 717%, 451%, 485%, 449% and 323% in the 55–59, 60–64, 65–69, 70–74, 75–79 and 80–84 age groups, respectively. The 45–49 age group increased to a peak in 2010–2014 and tended to plateau in 2015–2019. The 40–44 and 50–54 age groups showed a trend of rising to decline, with peaks occurring in 2010–2014. The 30–34 and 35–39 age groups showed a trend from rising to declining, with peaks occurring in 2005–2009. Figure 2b shows the age-specific rates of oral cancer in females. The age-specific rate of oral cancer in females stabilized in three age groups (30–34, 35–39 and 40–44) between 1980–1984 and 2015–2019. The age-specific rate of oral cancer in females in five age groups (45–49, 55–59, 65–69, 75–79 and 80–84) increased steadily between 1980–1984 and 2015–2019. The respective relative percent changes were 189%, 129%, 66%, 524% and 811%. The 50–54, 60–64, and 70–74 age groups showed a trend of rising to decline, with peaks occurring in 2010–2014.

### 3.3. Cohort-Specific Incidence Rate by Different Age Groups

Figure 3a shows the age-specific rates of men in different birth cohorts. The age-specific incidence rates in men increased with increasing birth years in most age groups. The 55–59 age group had a larger difference in incidence between 1925 and 1960. The incidence rate of males in the 55–59 age group was 12.6 per 100,000 in 1925 and 102.4 per 10,000 in 1960. The incidence of males born in 1960 was 8.12 times those born in 1925. The 30–34, 35–39, 40–44 and 55–59 age groups showed a trend of rising to decline, with peaks occurring in 1975, 1975, 1970 and 1960, respectively. Figure 3b shows the age-specific rates of women in different birth cohorts. The incidence rate in women in the 1900–1985 cohort was lower than that in men, and that in each age group showed a trend from rising to declining.

### 3.4. Age–Period–Cohort Model

Figure 4a shows the age effect of oral cancer by sex in Taiwan. By a fixed birth cohort, the incidence rate in both men and women increased gradually with age. For men, the incidence rate increased from 1.5 in the 32.5-year-old group to 88.3 in the 82.5-year-old group. For women, the incidence rate increased from 0.33 in the 32.5-year-old group to 23.8 in the 82.5-year-old group.

Figure 4b shows the period effect for oral cancer in Taiwan by sex. Based on the median of the fourth-period group (1995–2000), the relative risk of the different periods was observed. For men, the period effect in 2012–2017 was the strongest among all the periods, with a rate ratio of 1.8. For women, the period effect in 2012–2017 was also the strongest among all the periods, with a rate ratio of 1.6.

Figure 4c shows the cohort effect of oral cancer in Taiwan by sex. Based on the ninth cohort group (1940 cohort), a cohort effect was observed in the 1900 to 1985 cohort. In men, the rate ratio showed a trend of rising to decline, with peaks occurring in the 1975 cohort, with a rate ratio of 6.80. In women, the rate ratio showed a trend from rising to declining, with peaks occurring in the 1975 cohort, with a rate ratio of 2.76. The overall cohort effect of men was more obvious than that of women, and the fluctuation was larger.

### 3.5. Correlation of Risk Factors with Age-Standardized Incidence Rate of Oral Cancer

Figure 5a–c shows the correlations between the consumption of different risk factors and the incidence rate of oral cancer in men from 1980 to 2019. In Figure 5a, the consumption of tobacco products reached a peak in 2001, with an average value of 2070 for tobacco consumption per person. The trend of incidence of oral cancer was related to the changes in the consumption of cigarettes, with an r-value of 0.952 (*p* < 0.001) per Spearman’s correlation. In Figure 5b, the consumption of alcohol reached a peak in 1995, with an average of 39 L of alcohol consumption per person. The trend of incidence of oral cancer was related to the changes in the consumption of alcohol, with an r of 0.979 (*p* < 0.001) by Spearman’s correlation. As shown in Figure 5c, the production of betel nuts peaked in 1978, with an average 1.242-tonne betel nut yield per person. The trend in the incidence of oral cancer was related to the changes in the production of betel nut with an r-value of 0.963 (*p* < 0.001) per Spearman’s correlation. Regardless of the consumption change for any risk factor, the incidence rate of oral cancer in men increased from 1980 to a peak of 28.59 in 2014 and gradually tended to a plateau.

## 4. Discussion

In our study, from 1980 to 2019, the ASR of oral cancer increased from 4.19 to 27.19 per 100,000 population with an AAPC of 5.1% in men and from 1.16 to 2.8 per 100,000 population with an AAPC of 3.1% in women. The age-specific rate of oral cancer in males increased steadily between 1980–1984 and 2015–2019. The age-specific rate of oral cancer in females stabilized in three age groups (30–34, 35–39 and 40–44) and in five age groups (45–49, 55–59, 65–69, 75–79 and 80–84) increased steadily between 1980–1984 and 2015–2019. Age-specific incidence rates in men increased with the increasing birth year in most age groups. The incidence rate in women in the 1900–1985 cohort was lower than that in men, and that in each age group showed a trend of rising to decline. The Age–Period–Cohort model showed that the incidence rate in both men and women increased gradually with age in a fixed birth cohort. The period effect in 2012–2017 was the strongest among all the periods in both males and females. The overall cohort effect of men was more obvious than that of women, and the fluctuation was larger. Regardless of the consumption change for any risk factor, the incidence rate of oral cancer in men increased from 1980 to a peak of 28.59 in 2014 and gradually tended to a plateau.

A previous study based on the Global Burden of Disease database, which included 195 countries or territories, reported that the ASR of oral cancer in the world increased from 4.41 to 4.84 per 100,000 population from 1980 to 2017 [6]. Whether in Taiwan or the rest of the world, the incidence of oral cancer is increasing, but the increased rate of oral cancer incidence in Taiwan is faster than the global average. In addition, the incidence for males in 2019 was 27.19 per 100,000 and that for women was 2.82 per 10,000, which indicates that Taiwan is a high-risk region for oral cancer. In addition, previous research has reported that the incidence of oral cancer in men was two to three times greater than that in women [2], which means that men are more likely to suffer from oral cancer. The aforementioned finding is similar to that of our study, while the difference in incidence rates of oral cancer between males and females in Taiwan is larger than the global average.

A previous multicenter study including 6,151 cases of oral cancer reported that the average age of patients with oral cancer was 58.37 years, and the standard deviation of that was 15.77 years. Most of the cases (81.26%) occurred in the fifth to the eighth decades of life [16], which was similar to the pattern in Taiwan. In this study, the incidence rate increased apparently after the age was above 50 years. We found that the age-adjusted incidence had an increasing tendency with time, and the trend remained similar, even after stratification by age.

Oral cancer in its early stages is often asymptomatic. It is essential for clinicians to pay attention to precancerous lesions, such as leukoplakia and erythroplakia. Non-healing ulcers may appear as the malignancy develops. In the later stages of oral cancer, patients may suffer from dental loosening, dysphagia, neck masses and odynophagia [17]. Oral carcinogenesis is a multifactorial process that happens when epithelial cells are influenced by a change in genetic information, including disorders on TP53, EGFR (epidermal growth factor receptor), NOTCH1 (Notch homolog one gene, which are translocation-associated (Drosophila)), Cyclin D1, STAT3 (signal transducer and activator of transcription 3), CDKN2A (cyclin-dependent kinase inhibitor 2a), and Rb [2]. Due to mutations, exposure to biological factors or errors in DNA repair, keratinocytes may be unstable and result in malignant neoplastic changes [18]. Oral cancer has lots of risk factors, including tobacco, betel quid consumption, alcohol, diet, nutrition, mouthwash use and Maté consumption [19], which may explain the increasing incidence of oral cancer. Genetic factors of oral cancers have interactions with environmental factors. Based on the multifactor dimensionality reduction approach, a previous study with 103 oral cancer cases and 98 controls in Taiwan reported DNA repair genes X-ray repair cross-complementing groups (XRCCs) 1–4 had a strong association with environmental factors, such as alcohol, smoking and betel quid consumption. The risk of the XRCC2 rs2040639 heterozygous variant increased in smokers (adjusted odds ratio (OR) = 3.7, 95% confidence interval, CI = 1.1–12.1) and in alcohol drinkers (adjusted OR = 5.7, 95% CI = 1.4–23.2). For the gene–environment interaction, the estimated OR of oral cancer from drinking–betel quid was 32.9 (95% CI = 14.1–76.9), XRCC1–XRCC2–betel quid was 31.0 (95% CI = 14.0–64.7), XRCC1–XRCC2–age–betel quid was 49.8 (95% CI = 21.0–117.7) and XRCC1–XRCC2–age–drinking–betel quid was 82.9 (95% CI = 31.0–221.5) [20]. Nonetheless, due to the complexity of risk factors, it is difficult to explain the increasing incidence of oral cancer in Taiwan with a single risk factor. Among all risk factors, smoking and alcohol consumption were considered to be the main risk factors for oral cancer; they were present in 90% of cases [21], and they seemed synergistic with each other [22].

One meta-analysis that included 254 studies reported that smokers had a greater risk of developing oral cancer than non-smokers (relative risk, RR = 3.43; 95% CI = 2.37–4.94) [23]. Compared to current smokers, quitting smoking for 1–4 years could reduce the risk of head and neck cancer (OR = 0.70; 95% CI = 0.61–0.81). People who quit smoking for over 20 years can reduce the risk of head and neck cancer (OR = 0.23; 95% CI = 0.18–0.31) to a level equal to those who have never been smokers [24]. Tobacco consumption makes the oral epithelial cell be exposed to the reactive oxygen species, which has an impact on antioxidant defense mechanisms and promotes carcinogenesis [25]. In our study, the consumption of tobacco products reached a peak in 2001, with an average value of 2070 for tobacco consumption per person. However, the incidence rate of oral cancer in men increased from 1980, with a peak of 28.59 in 2014, and gradually tended toward a plateau. In Taiwan, the prevalence of smoking is high. From 1986 to 1990, the smoking rate among adult men increased from 59% to 63% due to Taiwan’s cigarette market opening up to foreign imports [26]. In 2001, the National Health Interview Survey of Taiwan reported that the prevalence was 46.8% smokers and 6.8% ex-smokers in adult males and 4.3% smokers and 0.5% ex-smokers in adult females [27]. Taiwan’s government established an anti-smoking law in 2009. Unfortunately, a cross-sectional study of 961 adults reported that 42% of sampled Taiwanese adults had smoked after the new law had been implemented [28]. Smoking seemed to have a cohort effect on oral cancer, and the high prevalence of smoking in an early cohort may explain the high incidence and difference between men and women regarding oral cancer in Taiwan.

A previous meta-analysis that included 17,085 cases of oral and pharyngeal cancers (OPC) reported that light drinkers (< or =1 drink per day) had a small risk of developing OPC (RR = 1.21, 95% CI = 1.10–1.33), and heavy drinkers (> or =4 drinks per day) had a higher risk of developing OPC (RR = 5.24, 95% CI = 4.36–6.30) compared to non- or occasional drinkers [29]. Alcohol could increase the permeability of the oral mucosa, dissolving epithelial lipid components, which may result in epithelial atrophy and interrupt the process of DNA synthesis and repair [30]. Acetaldehyde, which was metabolic from alcohol, has toxicity and is a cancer-causing component. In addition, the carcinogenesis of alcohol may interact with other factors, such as diet, smoking, and comorbidities [31]. Our study reported that the consumption of alcohol reached a peak in 1995, with an average of 39 L of alcohol consumption per person, while the incidence rate of oral cancer in men increased from 1980 with a peak of 28.59 in 2014 and gradually tended to a plateau. A study based on the Northeastern Taiwan Community Medicine Research Cohort, which included 3,387 participants, reported that 35.9% of participants were low-risk drinkers, and 9.1% of them were hazardous drinkers. In comparison with women, men had a higher frequency of drinking and higher total alcohol consumption amounts [32]. In Taiwan, a national survey that included 17,837 participants in 2014 and 18,626 participants in 2018 reported that alcohol-use behaviors in men had decreased significantly (with a decreasing prevalence of past-month alcohol use of 3.79%, binge drinking of 1.59%, and harmful alcohol use of 2.60%), while that in women increased significantly from 1.32% to 1.72% in harmful alcohol use [33]. Alcohol consumption seemed to have a cohort effect on oral cancer, and the difference in the trend between men and women may explain the sex differences in oral cancer in Taiwan.

Areca nut is an established cause of leukoplakia, submucous fibrosis and oral cancer [34]. A previous meta-analysis, including 50 reports, explored the relationship between chewing betel quid and the risk of oral cancer. In the Indian subcontinent, the meta-relative risk (mRR) for oral/oropharyngeal cancer was 2.56 (95% CI = 2.00–3.28) for betel quid consumption without added tobacco and 7.74 (95% CI = 5.38–11.13) for betel quid with added tobacco. In the Taiwan subcontinent, the mRR for oral/oropharyngeal cancer was 10.98 (95% CI = 4.86–24.84) for betel quid without added tobacco [35]. Arecoline, which is contained in areca nut, may induce DNA damage, which results in oral carcinogenesis [36]. The possible effect of this is that arecoline could downregulate and inhibit the p53 expression [37]. Oral submucous fibrosis is a precancerous lesion that has individual susceptibility related to betel quid consumption due to cumulative exposure. Based on PCR-based restriction fragment length polymorphism assays, a comparative study with 166 patients with oral submucous fibrosis and 284 betel quid chewers reported the highest genotypes of oral submucous fibrosis risk for collagen 1A1, collagen 1A2, collagenase-1, transforming growth factor beta1, lysyl oxidase and cystatin C were CC, AA, TT, CC, AA and AA, respectively, in the low-exposure group. In the high-exposure group, the highest genotypes of oral submucous fibrosis risk for collagen 1A1, collagen 1A2, collagenase-1, transforming growth factor beta1, lysyl oxidase and cystatin C were TT, BB, AA, CC, GG and AA, respectively [38]. According to our study, the production of betel nut peaked in 1978, with an average of 1.242 tonnes of betel nut yield per person, while the incidence rate of oral cancer in men increased from 1980 with a peak of 28.59 in 2014 and gradually tended toward a plateau. In Asia, betel nut products can be classified as either with or without tobacco [39]. In Taiwan, betel nut without tobacco is more common than with it [40]. A previous meta-analysis including 19 studies in South Asia and the Pacific reported that betel quid without tobacco also has a serious impact on the risk of oral cancer (OR = 2.82, 95% CI = 2.35–3.40) [41]. In Taiwan, the prevalence of betel nut chewing among people over 15 years of age was about 8.8–16.1%, and the prevalence in males was larger than in females. A previous comparative study reported that the annual average betel nut consumption per person could account for 86.2% of the variation in the incidence of oral cavity cancer by multiple regression models [42]. Betel quid consumption seemed to have a cohort effect on oral cancer, and the difference in the trend between men and women may explain the sex differences for oral cancer in Taiwan.

The importance of our study lies in the fact that we portrayed the long-term trends of oral cancer in Taiwan from 1980 to 2019, and we used an age–period–cohort model to analyze it. Regarding the consumption of alcohol and cigarettes and the production of betel quid, our study found that the production of betel quid varied widely. Thus, we assumed that the change in the consumption behavior of betel quid was the major cause of the change in the trend of oral cancer in Taiwan. A previous study based on TCR reported that over 80% of men with head and neck cancer had ever smoked, over 70% of them had ever drunk, and over 60% of patients with head and neck cancer had ever chewed betel quid. Additionally, over 70% of patients with oral cancer had ever chewed betel quid [43]. A previous multi-center study with 921 patients with head and neck cancer and 806 controls in East Asia reported that the risk of head and neck cancer would be elevated by tobacco (OR = 1.58, 95%CI = 1.18–2.13), alcohol (OR = 2.29, 95%CI = 1.76–2.99) and betel quid (OR = 8.23, 95%CI = 5.31–12.75). Tobacco and/or alcohol consumption accounts for 47.2% of head and neck cancer, and betel quid chewing alone accounts for 28.7% [44]. Therefore, we assumed that betel quid was a major risk factor for oral cancer in the East Asia region.

According to the Global Cancer Statistics 2020, the age-standardized mortality rate for cancer of the lip and oral cavity was 2.8 per 100,000 men and 1.0 per 100,000 women worldwide [1]. The age-standardized mortality rates of oral cancer increased from 10.14 per 100,000 in 1991 to 23.39 per 100,000 people in 1999 in Taiwanese men between the ages of 30 and 64. In 1999, Taiwan implemented a nationwide screening program for oral cancer. The program includes selective screening and a free oral mucosal examination every two years for individuals over 30 years of age who engage in high-risk behaviors, such as tobacco smoking and betel nut chewing [45]. In 1997, Taiwan’s government implemented the Tobacco Hazards Prevention Act, which prohibited the purchase of cigarettes by teenagers under 18 years of age and pregnant women and limited smoking in public environments [46]. Furthermore, the tobacco health and welfare surcharges were increased to 5, 10 and 20 New Taiwan Dollars (NTD) per pack in 2002, 2006 and 2009, respectively. Additionally, the tobacco tax was raised to NTD 20 per pack in 2017 [47]. In 2008, Taiwan’s government implemented a new regulation that aimed to “eliminate betel nut farms and repurpose the land”, but the outcomes were only moderate. In 2014, another document called the “Management Plan Concerning the Betel Nut” was introduced, with the aim of eliminating 10,000 hectares of betel nut farms over the next three years [48]. Currently, the Taiwanese government levies an excise tax of NTD 26 on each liter of beer and NTD 2.5 per liter of distilled liquor for each percentage of alcohol content. Reprocessed alcoholic beverages are taxed at NTD 7 for each percent of alcohol when the alcohol content is less than 20% by volume and NTD 185 per liter when the alcohol content is over 20% [49]. Overall, the Taiwanese government attaches great importance to the dangerous factors associated with oral cancer and has implemented several policies to address them.

The study has some limitations. As the other database found, a portion of cases potentially did not receive pathological verification. Nevertheless, TCR’s morphological verification percentage was 93.0%, and the integrity was 98.4% [11], minimizing the errors of cancer misdiagnosis. In addition, the Taiwanese government established a national oral cancer screening program in 1999, which may have increased the diagnosis of oral cancer in its early stages in recent years [50]. Moreover, our study lacks individual cancer risk factor data, which may result in an ecological fallacy between oral cancer and related risk factors. Additionally, regarding the aforementioned research, there were a lot of risk factors for oral cancer, while other potential risk factors were discussed in our study. However, the importance of alcohol, betel quid and smoking were greater than the others. Finally, we did not include all subsites of oral cancer in the analysis. Nonetheless, given the small percentage of people with floor-of-the-mouth or gum cancer, changes in their incidence will not affect the overall trend of the incidence rate of oral cancer, which is what we focused on in this study. We strongly suggest avoiding the risk factors to prevent oral cancer, particularly smoking, alcohol and betel quid consumption. Among all the risk factors, betel quid consumption was relatively important in Asia. To confirm our study’s findings, further large and long-term cohort studies are needed.

## 5. Conclusions

In summary, we herein describe the characteristics of oral cancer in Taiwan. The incidence of oral cancer is increasing. A cohort effect of oral cancer was observed, which may be due to the consumption of cigarettes, alcohol and betel quid. We suggest that the Taiwan government attach more importance to the risk factors for oral cancers, such as cigarettes, alcohol and betel quid, and there is a need for policies aimed at preventing oral cancer. To confirm the independent and/or interactive effects of the risk factors, further large cohort studies are necessary.

## Figures and Tables

**Figure 1 cancers-15-02175-f001:**
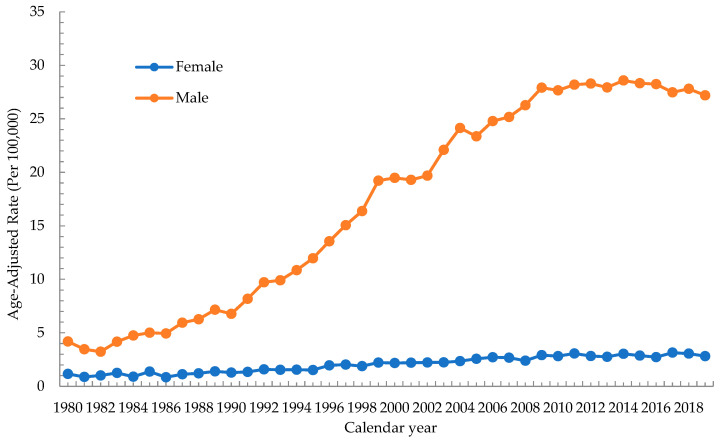
Secular trend in age-standardized incidence rates of oral cancer in Taiwan, 1980–2019.

**Figure 2 cancers-15-02175-f002:**
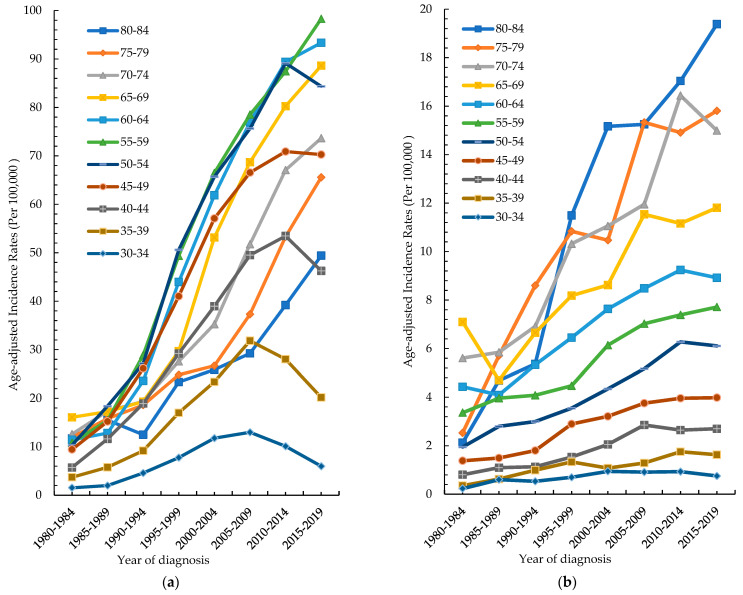
Age-specific incidence rates of oral cancer by year of diagnosis: (**a**) male, (**b**) females.

**Figure 3 cancers-15-02175-f003:**
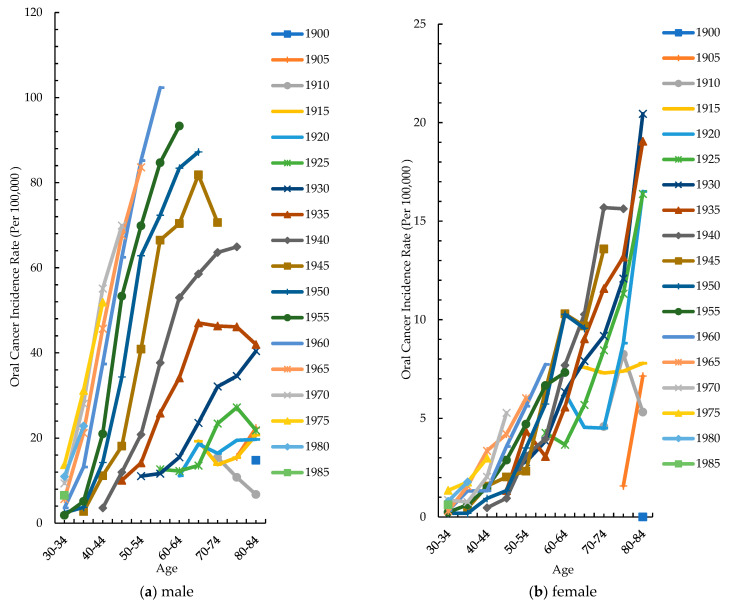
Age-specific incidence rates of oral cancer by birth year: (**a**) males, (**b**) females.

**Figure 4 cancers-15-02175-f004:**
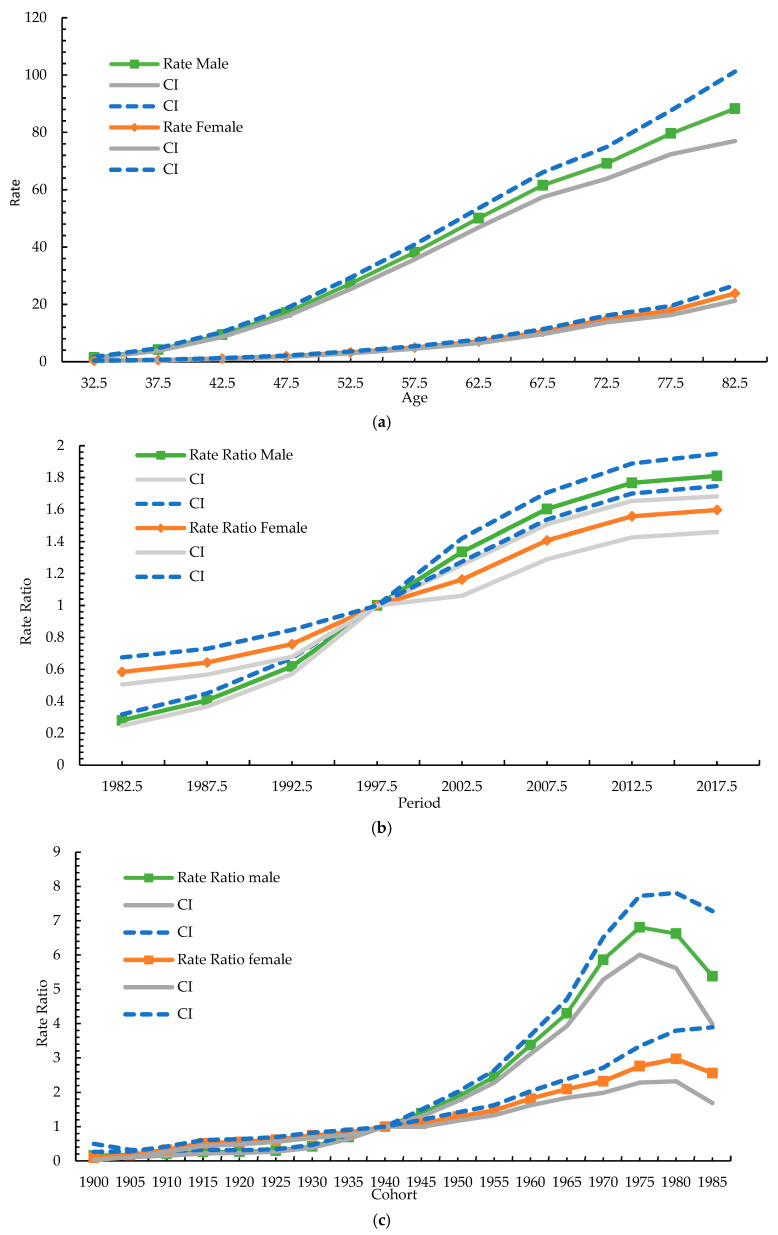
(**a**) Age effect of oral cancer in Taiwan by sex. Age: 32.5–82.5; (**b**) Period effect of oral cancer in Taiwan by sex. Period: 1980–2019; (**c**) Cohort effect of oral cancer in Taiwan by sex. Cohort: 1900–1985.

**Figure 5 cancers-15-02175-f005:**
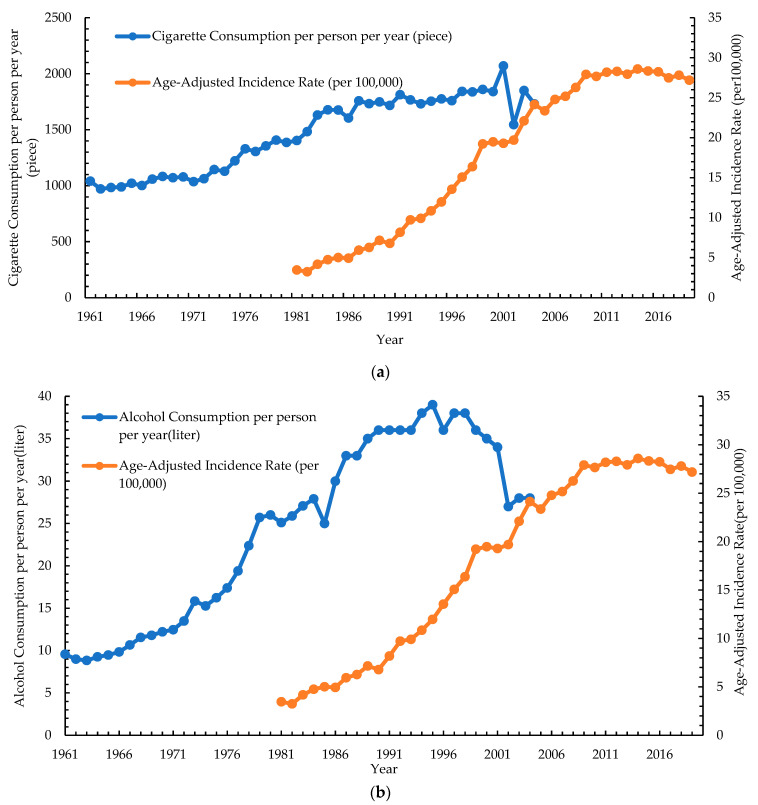
Association between the consumption/production risk factors and age-adjusted incidence rate of oral cancer: (**a**) Cigarette consumption per person per year (piece). (**b**) Alcohol consumption per person per year (liter). (**c**) The yield of betel nut/population (tonne per person).

## Data Availability

Data derived from public domain resources.

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
