# Peer review of "Epidemiology of Oral Cancer in Taiwan: A Population-Based Cancer Registry Study"

_cancers, 2023, doi:10.3390/cancers15072175_

Round 1

Reviewer 1 Report

Dear Authors,

I read the paper with a big interest. It is well written.

I have a couple of minor comments.

Line 20-21 Please, paraphrase the sentence. “….this trend may be related to….” because it is a discussion or speculation.

Same with lines 22-25.

Moreover, “further long-term cohort study is (not was)”.

Line 33 double space between “by and Europe”.

Figure 4a 4b and 4c it is enough if you will use the CI acronym for both lines, so please delete CIHi CiLO and keep grey line for one CI, and blue dots for another CI. CI for itself means low and high bound.

All the best

Author Response

Response to Reviewer 1 Comments

Point 1:

I read the paper with a big interest. It is well written.

I have a couple of minor comments.

Line 20-21 Please, paraphrase the sentence. “….this trend may be related to….” because it is a discussion or speculation.

Response 1:

Thank you for your suggestion and comments. We adjusted the results of description. It was revised in the new manuscript (page 1, line 31-33).

“The trend of incidence of oral cancer was related to changes in the consumption of cigarettes and alcohol and production of betel quid, with r values of 0.952, 0.979 and 0.963, respectively (all p values <0.001).”

Point 2:

Same with lines 22-25.

Response 2:

Thank you for your suggestion and comments. We adjusted the results of description. It was revised in the new manuscript (page 1, line 31-33).

“The trend of incidence of oral cancer was related to changes in the consumption of cigarettes and alcohol and production of betel quid, with r values of 0.952, 0.979 and 0.963, respectively (all p values <0.001).”

Point 3:

Moreover, “further long-term cohort study is (not was)”.

Response 3:

Thank you for your careful review. We appreciate your effort in addressing the grammatical error, and we have removed this sentence. The last sentence of the abstract in mentioned in page 1, line 33-34.

“We strongly suggest avoiding these risk factors in order to prevent OC.”

In addition, in accordance with the reviewer's suggestion, we have availed the services of an English editing service to ensure the accuracy and clarity of the manuscript.

Point 4:

Line 33 double space between “by and Europe”.

Response 4:

Thank you for your careful review. We appreciate your effort in addressing the grammatical error, and we have noted the revision made in the new manuscript (page 1, line 41-42).

Point 5:

Figure 4a 4b and 4c it is enough if you will use the CI acronym for both lines, so please delete CIHi CiLO and keep grey line for one CI, and blue dots for another CI. CI for itself means low and high bound.

Response 5:

Thank you for your careful review and insightful comments. We have made the necessary adjustments to Figure 4a, 4b, and 4c based on your suggestion to improve their comprehensibility.

Reviewer 2 Report

General Comments:

  • The manuscript provides a good overview of the characteristics of oral cancer in Taiwan.
  • In terms of English quality, the manuscript is generally understandable but there are several instances where the language could be improved to enhance clarity and readability. Some sentences are too long and complex, which can make it difficult for readers to follow the logic of the argument. Additionally, there are some grammatical errors and awkward phrasings that could be corrected. I would recommend that the authors have a native English speaker review the manuscript or consider using a language editing service to ensure that the language is polished and easy to read. For example,
  1. "Nowadays" is used frequently throughout the manuscript, which is not considered formal English. Instead, "currently" or "presently" could be used.
  2. "This research was aimed to" is not grammatically correct. Instead, it should be "This research aimed to" or "The aim of this research was to".
  3. There are instances of awkward or unclear phrasing, such as "the association between cigarette smoking, alcohol drinking and betel nut chewing with oral cancer is becoming more and more apparent" which could be revised to "the association between cigarette smoking, alcohol drinking, and betel nut chewing with oral cancer is increasingly apparent".

Specific Comments:

Title:

  • The title could be more specific and concise to accurately reflect the content of the manuscript. It could benefit from a more specific description of the focus of the research, such as "Epidemiology of Oral Cancer in Taiwan: A Retrospective Cohort Study."

Abstract:

  • The abstract provides a good overview of the study; however, it could be improved by providing more specific information about the study population, such as age range and demographic characteristics, and by briefly summarizing the most significant findings.

Introduction:

  • Provide a more comprehensive and up-to-date overview of the prevalence, risk factors, and consequences of oral cancer in Taiwan.
  • The research question and objectives of the study not clearly stated
  • Provide a more detailed rationale for the study and its significance.

Methods:

  • The methods section is lack of the study design, sampling strategy, and data collection procedures.
  • Why did you choose data until 2019? Its already 2023, data should be until 2022.
  • The statistical methods need to be clearly described and justified.
  • Provide more information on how did you assess the validity and reliability of the data.

Results:

  • The results section is well-organized and provides detailed information about the findings of the study. However, it could benefit from a more thorough discussion of the statistical significance of the results, as well as an explanation of any unexpected or conflicting findings.

Discussion:

  • The discussion section should provide a more in-depth and critical analysis of the study findings.
  • The authors should relate their findings to the existing literature and highlight their study's unique contributions to the field.
  • The limitations of the study should be acknowledged, and future research directions should be suggested.

Conclusion:

  • The conclusion provides a brief summary of the study's findings and recommendations for future research. However, it could be improved by providing more specific information about the potential implications of the findings for public health in Taiwan, as well as the specific areas in which further research is needed.

Overall, the manuscript has the potential to make a valuable contribution to the literature on oral cancer in Taiwan. However, the authors need to address the above-mentioned issues to improve the clarity, coherence, and rigor of the manuscript.

Author Response

Response to Reviewer 2 Comments

Point 1:

The manuscript provides a good overview of the characteristics of oral cancer in Taiwan.

In terms of English quality, the manuscript is generally understandable but there are several instances where the language could be improved to enhance clarity and readability.

Some sentences are too long and complex, which can make it difficult for readers to follow the logic of the argument.

Additionally, there are some grammatical errors and awkward phrasings that could be corrected. I would recommend that the authors have a native English speaker review the manuscript or consider using a language editing service to ensure that the language is polished and easy to read. For example,

  1. "Nowadays" is used frequently throughout the manuscript, which is not considered formal English. Instead, "currently" or "presently" could be used.
  2. "This research was aimed to" is not grammatically correct. Instead, it should be "This research aimed to" or "The aim of this research was to".
  3. There are instances of awkward or unclear phrasing, such as "the association between cigarette smoking, alcohol drinking and betel nut chewing with oral cancer is becoming more and more apparent" which could be revised to "the association between cigarette smoking, alcohol drinking, and betel nut chewing with oral cancer is increasingly apparent".

Response 1:

Thank you for your suggestions and comments. We have employed a language editing service to refine our manuscript, and we are pleased to inform you that the latest version of our manuscript is now highly readable.

Point 2:

Specific Comments:

Title:

The title could be more specific and concise to accurately reflect the content of the manuscript. It could benefit from a more specific description of the focus of the research, such as "Epidemiology of Oral Cancer in Taiwan: A Retrospective Cohort

Study."

Response 2:

Thank you for your suggestion and comments. We have revised the title of our research to "Epidemiology of Oral Cancer in Taiwan: A Population-based Cancer Registry Study" to describe the focus of our research more accurately (page 1, line 2-3).

Point 3:

Abstract:

The abstract provides a good overview of the study; however, it could be improved by providing more specific information about the study population, such as age range and demographic characteristics, and by briefly summarizing the most significant findings.

Response 3:

Thank you for your valuable comments and input. It was revised in the new manuscript (page 1, line 22-34) to provide a more specific summary of this study.

“Abstract: Oral cancer (OC) is one of the most common cancers worldwide, and its incidence has regional differences. In this study, the cancer registry database obtained from 1980 to 2019 was used to analyze the characteristic of incidence of OC by average annual percentage change (AAPC) and an age–period–cohort model. Spearman’s correlation was used to analyze the relationship between the age-standard incidence rates (ASR) of OC and related risk factors. Our results showed that the ASR of OC increased from 4.19 to 27.19 per 100,000 population with an AAPC of 5.1% (95% CI=3.9-6.3, p value<0.001) in men and from 1.16 to 2.8 per 100,000 population with an AAPC of 3.1% (95% CI=2.6-3.6, p value<0.001) in women between 1980-1984 and 2015-2019. The age–period–cohort model reported a trend of rising then declining for the rate ratio in men, with peaks occurring in the 1975 cohort, with a rate ratio of 6.80. The trend of incidence of oral cancer was related to changes in the consumption of cigarettes and alcohol and production of betel quid, with r values of 0.952, 0.979 and 0.963, respectively (all p values <0.001). We strongly suggest avoiding these risk factors in order to prevent OC.”

Point 4:

Introduction:

Provide a more comprehensive and up-to-date overview of the prevalence, risk factors, and consequences of oral cancer in Taiwan.

Response 4:

Thanks for your constructive feedback and suggestions. We have revised the new manuscript to include more specific information on the epidemiology of oral cancer in Taiwan. It was mentioned in the new manuscript (page 1, line 45-47).

“Oral cancer has multiple risk factors, including smoking, other tobacco consumption, snuff dipping, alcohol, sunlight exposure, viruses, and so on [3].

It was revised in the new manuscript (page 2, line 52-66).

Based on the Global Cancer Statistics 2020, the age-standardized mortality of cancer of the lip and oral cavity was 2.8 per 100,000 in men and 1.0 per 100,000 in women worldwide [1, 3, 6]. In many countries, including the USA, population-based screening for oral cancer is not recommended due to insufficient evidence demonstrating its efficacy in reducing mortality[7]. Taiwan has been conducting a national population-based screening program for oral cancer since 2004. From 2004 to 2009, the screening rate was 55.1%, and mortality decreased by 26% in the screened group[8]. The epidemiology of oral cancer and the relationship between oral cancer and its risk factors may differ in the different countries and at different times. For example, betel leaf and areca nut consumption are common social practices in South Asia, Southeast Asia, and Pacific Asia, as well as in emigrated communities in North America and Europe, and this has been identified as a risk factor for head and neck cancers[9]”

  1. Sung, H.; Ferlay, J.; Siegel, R. L.; Laversanne, M.; Soerjomataram, I.; Jemal, A.; Bray, F., Global Cancer Statistics 2020: GLOBOCAN Estimates of Incidence and Mortality Worldwide for 36 Cancers in 185 Countries. CA Cancer J Clin 2021, 71, (3), 209-249.
  2. Warnakulasuriya, S., Global epidemiology of oral and oropharyngeal cancer. Oral Oncol 2009, 45, (4-5), 309-16.
  3. Ren, Z. H.; Hu, C. Y.; He, H. R.; Li, Y. J.; Lyu, J., Global and regional burdens of oral cancer from 1990 to 2017: Results from the global burden of disease study. Cancer Commun (Lond) 2020, 40, (2-3), 81-92.
  4. Warnakulasuriya, S.; Kerr, A. R., Oral Cancer Screening: Past, Present, and Future. J Dent Res 2021, 100, (12), 1313-1320.
  5. Chuang, S. L.; Su, W. W.; Chen, S. L.; Yen, A. M.; Wang, C. P.; Fann, J. C.; Chiu, S. Y.; Lee, Y. C.; Chiu, H. M.; Chang, D. C.; Jou, Y. Y.; Wu, C. Y.; Chen, H. H.; Chen, M. K.; Chiou, S. T., Population-based screening program for reducing oral cancer mortality in 2,334,299 Taiwanese cigarette smokers and/or betel quid chewers. Cancer 2017, 123, (9), 1597-1609.
  6. Cohen, N.; Fedewa, S.; Chen, A. Y., Epidemiology and Demographics of the Head and Neck Cancer Population. Oral Maxillofac Surg Clin North Am 2018, 30, (4), 381-395.

Point 5:

The research question and objectives of the study not clearly stated.

Provide a more detailed rationale for the study and its significance.

Response 5:

Appreciate your careful review and insights. We have revised the new manuscript not only followed the comment 4 “Provide a more comprehensive and up-to-date overview of the prevalence, risk factors, and consequences of oral cancer in Taiwan.”; but also to include a more detailed rationale and significance of this study (page 2, line 63-66). “Due to the serious threat of oral cancer towards public health, this study aimed to offer the latest information regarding public health based on a population-based database to analyze the age–period–cohort (APC) effect of oral cancer.”

Point 6:

Methods:

The methods section is lack of the study design, sampling strategy, and data collection procedures.

Response 6:

Thanks for your valuable input and suggestions. It was revised in the new manuscript (page 2, line 69-73) to provide a more specific description of the methods used in this study.

“This research was an observational study. The number of newly diagnosed oral cancer cases in each age group (0–89 years) from 1980 to 2019 was retrieved from the Taiwan Cancer Registry (TCR), a nationwide population-based cancer registry system maintained by the Health Promotion Administration, and all data in this study were obtained from the public website without personal information [10].”

  1. Health Promotion Administration Taiwan Cancer Registry Database. https://cris.hpa.gov.tw/pagepub/Home.aspx (Nov 11),

Point 7:

Why did you choose data until 2019? Its already 2023, data should be until 2022.

Response 7:

Thank you for your suggestion. The most recent public statistics on oral cancer from Taiwan cancer registry in Taiwan, updated within the last two or three months to include data from 2020. We have observed data from 2019 and 2020, and the prevalence of oral cancer in 2020 was comparable to that of 2019. As our result, the epidemiology of oral cancer has remained stable since 2008. Therefore, we did not include the data from 2019 to 2022.

Point 8:

The statistical methods need to be clearly described and justified.

Response 8:

Thank you for your suggestion and comments. The statistical method was mentioned more clearly in the new manuscript (page 2, line 95-96, page3, line 97-104).

“To assess the secular trend in oral cancer incidence, the study utilized the National Cancer Institute’s online tool to calculate the average annual percentage change (AAPC) [16]. Additionally, the study investigated the period and cohort effects on oral cancer incidence through age–period–cohort modeling using the same web tool. This approach provided the rate ratios (RRs) that compared the incidence of oral cancers in different periods (period effects) and cohorts (cohorts) to the reference points [16]. The relationship between ASR of oral cancer and related risk factors was analyzed using Spearman’s correlation and statistical analysis was conducted by Microsoft™ Excel™ 365 MSO 16.0.13528.203018 64bit (Microsoft Corporation, Redmond, Washington, USA).”

  1. Dhanuthai, K.; Rojanawatsirivej, S.; Thosaporn, W.; Kintarak, S.; Subarnbhesaj, A.; Darling, M.; Kryshtalskyj, E.; Chiang, C. P.; Shin, H. I.; Choi, S. Y.; Lee, S. S.; Aminishakib, P., Oral cancer: A multicenter study. Med Oral Patol Oral Cir Bucal 2018, 23, (1), e23-e29.

Point 9:

Provide more information on how did you assess the validity and reliability of the data.

Response 9:

Thank you for your suggestion and comments. We used DCO% and morphological verification percentage (MV%) to morphological verification percentage (MV%), and it was revised in the new manuscript (page 2, line 81-83).

“The percentage of death certificate only (DCO%) of all cancers decreased from 8.84% in 1998 to 0.71% in 2019, and the morphological verification percentage (MV%) in 2019 was 93.47% (men: 92.31%, women: 94.77%)[11].”

  1. Chiang, C. J.; Wang, Y. W.; Lee, W. C., Taiwan's Nationwide Cancer Registry System of 40 years: Past, present, and fu-ture. J Formos Med Assoc 2019, 118, (5), 856-858.

Point 10:

Results:

The results section is well-organized and provides detailed information about the findings of the study. However, it could benefit from a more thorough discussion of the statistical significance of the results, as well as an explanation of any unexpected or conflicting findings.

Response 10:

Thank you so much for your praise.

Point 11:

Discussion:

The discussion section should provide a more in-depth and critical analysis of the study findings.

The authors should relate their findings to the existing literature and highlight their study's unique contributions to the field. The limitations of the study should be acknowledged, and future research directions should be suggested.

Response 11:

Thank you for your suggestion and comments. In the new manuscript, we have included a detailed analysis of our study findings in order to provide deeper insights and enhance the overall significance of our study (page 11, line 354-366, and page 12, line 367-377).

“According to the Global Cancer Statistics 2020, the age-standardized mortality rate for cancer of the lip and oral cavity was 2.8 per 100,000 men and 1.0 per 100,000 women worldwide[1]. The age-standardized mortality rates of oral cancer increased from 10.14 per 100,000 in 1991 to 23.39 per 100,000 people in 1999 in Taiwanese men between the ages of 30 and 64. In 1999, Taiwan implemented a nationwide screening program for oral cancer. The program includes selective screening and a free oral mucosal examination every two years for individuals over 30 years of age who engage in high-risk behaviors such as tobacco smoking and betel nut chewing[45]. In 1997, Taiwan’s government implemented the Tobacco Hazards Prevention Act, which prohibited the purchase of cigarettes by teenagers under 18 years of age and pregnant women and limited smoking in public environments[46]. Furthermore, the tobacco health and welfare surcharges were increased to 5, 10 and 20 New Taiwan Dollars (NTD) per pack in 2002, 2006 and 2009, respectively. Additionally, the tobacco tax was raised to NTD 20 per pack in 2017[47]. In 2008, Taiwan’s government implemented a new regulation that aimed to "eliminate betel nut farms and repurpose the land," but the outcomes were only moderate. In 2014, another document called the "Management Plan Concerning the Betel Nut" was introduced, with the aim of eliminating 10,000 hectares of betel nut farms over the next three years[48]. Currently, the Taiwanese government levies an excise tax of NTD 26 on each liter of beer and NTD 2.5 per liter of distilled liquor for each percentage of alcohol content. Reprocessed alcoholic beverages are taxed at NTD 7 for each percent of alcohol when the alcohol content is less than 20% by volume, and NTD 185 per liter when the alcohol content is over 20%[49]. Overall, the Taiwanese government attaches great importance to the dangerous factors associated with oral cancer and has implemented several policies to address them.”

  1. Sung, H.; Ferlay, J.; Siegel, R. L.; Laversanne, M.; Soerjomataram, I.; Jemal, A.; Bray, F., Global Cancer Statistics 2020: GLOBOCAN Estimates of Incidence and Mortality Worldwide for 36 Cancers in 185 Countries. CA Cancer J Clin 2021, 71, (3), 209-249.
  2. Su, S. Y., Evaluation of Nationwide Oral Mucosal Screening Program for Oral Cancer Mortality among Men in Taiwan. Int J Environ Res Public Health 2022, 19, (21).
  3. Ministry of Health and Welfare Tobacco Hazards Prevention Act ,. https://law.moj.gov.tw/ENG/LawClass/LawAll.aspx?pcode=L0070021 (Mar 22),
  4. Ministry of Finance Tobacco and Alcohol Tax Act ,. https://law.moj.gov.tw/ENG/LawClass/LawAll.aspx?pcode=G0330010 (Mar 22),
  5. Tham, J.; Sem, G.; Sit, E.; Tai, M. C., The ethics of betel nut consumption in Taiwan. J Med Ethics 2017, 43, (11), 739-740.
  6. Yeh, C. Y.; Ho, L. M.; Lee, J. M.; Hwang, J. Y., The possible impact of an alcohol welfare surcharge on consumption of alcoholic beverages in Taiwan. BMC Public Health 2013, 13, 810.

Point 12:

Conclusion:

The conclusion provides a brief summary of the study's findings and recommendations for future research. However, it could be improved by providing more specific information about the potential implications of the findings for public health in Taiwan, as well as the specific areas in which further research is needed.

Overall, the manuscript has the potential to make a valuable contribution to the literature on oral cancer in Taiwan. However, the authors need to address the above-mentioned issues to improve the clarity, coherence, and rigor of the manuscript.

Response 12:

Thankful for your thoughtful comments and suggestions. We have revised the new manuscript (page 12, line 396-402) to provide more detailed information about the potential implications of the findings for public health in Taiwan.

“In summary, we herein describe the characteristics of oral cancer in Taiwan. The incidence of oral cancer is increasing. A cohort effect of oral cancer was observed, which may be due to the consumption of cigarettes, alcohol and betel quid. We suggest that the Taiwan government attach more importance to the risk factors for oral cancers, such as cigarettes, alcohol and betel quid, and there is a need for policies aimed at preventing oral cancer. To confirm the independent and/or interactive effects of the risk factors, further large cohort studies are necessary.

Thanks for your comments and suggestion.”

Reviewer 3 Report

Very Respected Authors,

After carefully reading the manuscript I have few sugesstions. Conclusion in the Abstract has to be in agreement with the findings and with the objective of the manuscript. The Introduction of the manuscript could be longer and with more relevant data. The references could be more recent.

Author Response

Response to Reviewer 3 Comments

Point 1:

After carefully reading the manuscript I have few sugesstions. Conclusion in the abstract has to be in agreement with the findings and with the objective of the manuscript.

Response 1:

Thank you for your suggestion and comments. It was revised in the new manuscript (page 1, line 22-34) to s provide a more concise and specific summary of the study.

“Abstract: Oral cancer (OC) is one of the most common cancers worldwide, and its incidence has regional differences. In this study, the cancer registry database obtained from 1980 to 2019 was used to analyze the characteristic of incidence of OC by average annual percentage change (AAPC) and an age–period–cohort model. Spearman’s correlation was used to analyze the relationship between the age-standard incidence rates (ASR) of OC and related risk factors. Our results showed that the ASR of OC increased from 4.19 to 27.19 per 100,000 population with an AAPC of 5.1% (95% CI=3.9-6.3, p value<0.001) in men and from 1.16 to 2.8 per 100,000 population with an AAPC of 3.1% (95% CI=2.6-3.6, p value<0.001) in women between 1980-1984 and 2015-2019. The age–period–cohort model reported a trend of rising then declining for the rate ratio in men, with peaks occurring in the 1975 cohort, with a rate ratio of 6.80. The trend of incidence of oral cancer was related to changes in the consumption of cigarettes and alcohol and production of betel quid, with r values of 0.952, 0.979 and 0.963, respectively (all p values <0.001). We strongly suggest avoiding these risk factors in order to prevent OC.”

Point 2:

The Introduction of the manuscript could be longer and with more relevant data.

Response 2:

Thank you for your suggestion and comments. It was revised in the new manuscript (page 1, line 38-43, page 2, line 44-66) to provide more information of this study.

“Based on the Global Cancer Statistics 2020, oral cancer is one of the most common cancers worldwide. The global age-standardized rate (ASR) of oral cancer was 6.0 per 10,000 in males and 2.3 per 10,000 in females. However, the incidence of oral cancer has regional variation. Among the six continents, Asia had the highest incidence of oral cancer (65.8%), followed by Europe (17.3%) and North America (7.3%)[1]. According to the Human Development Index of the United Nations Development Program, the incidence of oral cancer is higher in countries with a better development index, while mortality is higher in less developed areas, which indicates social inequality [2]. Oral cancer has multiple risk factors, including smoking, other tobacco consumption, snuff dipping, alcohol, sunlight exposure, viruses, and so on [3]. The most common pathology of oral cancer is squamous cell carcinoma and the major risk factors are tobacco and alcohol[4]. A previous cohort study with 177,271 adult men reported that the mortality hazard ratio (HR) was 12.52 for oral cancer in chewers of betel quid [5]. Surgical resection with or without postoperative radiation or chemoradiation therapy is the standard treatment for management of oral cancer[4]. Based on the Global Cancer Statistics 2020, the age-standardized mortality of cancer of the lip and oral cavity was 2.8 per 100,000 in men and 1.0 per 100,000 in women worldwide [1, 3, 6]. In many countries, including the USA, population-based screening for oral cancer is not recommended due to insufficient evidence demonstrating its efficacy in reducing mortality[7]. Taiwan has been conducting a national population-based screening program for oral cancer since 2004. From 2004 to 2009, the screening rate was 55.1%, and mortality decreased by 26% in the screened group[8]. The epidemiology of oral cancer and the relationship between oral cancer and its risk factors may differ in the different countries and at different times. For example, betel leaf and areca nut consumption are common social practices in South Asia, Southeast Asia, and Pacific Asia, as well as in emigrated communities in North America and Europe, and this has been identified as a risk factor for head and neck cancers[9]. Due to the serious threat of oral cancer towards public health, this study aimed to offer the latest information regarding public health based on a population-based database to analyze the age–period–cohort (APC) effect of oral cancer.

  1. Sung, H.; Ferlay, J.; Siegel, R. L.; Laversanne, M.; Soerjomataram, I.; Jemal, A.; Bray, F., Global Cancer Statistics 2020: GLOBOCAN Estimates of Incidence and Mortality Worldwide for 36 Cancers in 185 Countries. CA Cancer J Clin 2021, 71, (3), 209-249.
  2. Rivera, C., Essentials of oral cancer. Int J Clin Exp Pathol 2015, 8, (9), 11884-94.
  3. Warnakulasuriya, S., Global epidemiology of oral and oropharyngeal cancer. Oral Oncol 2009, 45, (4-5), 309-16.
  4. Montero, P. H.; Patel, S. G., Cancer of the oral cavity. Surg Oncol Clin N Am 2015, 24, (3), 491-508.
  5. Wen, C. P.; Tsai, M. K.; Chung, W. S.; Hsu, H. L.; Chang, Y. C.; Chan, H. T.; Chiang, P. H.; Cheng, T. Y.; Tsai, S. P., Can-cer risks from betel quid chewing beyond oral cancer: a multiple-site carcinogen when acting with smoking. Cancer Causes Control 2010, 21, (9), 1427-35.
  6. Ren, Z. H.; Hu, C. Y.; He, H. R.; Li, Y. J.; Lyu, J., Global and regional burdens of oral cancer from 1990 to 2017: Results from the global burden of disease study. Cancer Commun (Lond) 2020, 40, (2-3), 81-92.
  7. Warnakulasuriya, S.; Kerr, A. R., Oral Cancer Screening: Past, Present, and Future. J Dent Res 2021, 100, (12), 1313-1320.
  8. Chuang, S. L.; Su, W. W.; Chen, S. L.; Yen, A. M.; Wang, C. P.; Fann, J. C.; Chiu, S. Y.; Lee, Y. C.; Chiu, H. M.; Chang, D. C.; Jou, Y. Y.; Wu, C. Y.; Chen, H. H.; Chen, M. K.; Chiou, S. T., Population-based screening program for reducing oral cancer mortality in 2,334,299 Taiwanese cigarette smokers and/or betel quid chewers. Cancer 2017, 123, (9), 1597-1609.
  9. Cohen, N.; Fedewa, S.; Chen, A. Y., Epidemiology and Demographics of the Head and Neck Cancer Population. Oral Maxillofac Surg Clin North Am 2018, 30, (4), 381-395.

Point 3:

The references could be more recent.

Response 3:

Grateful for your helpful feedback and suggestions. We have included the following new reference to provide more updated information on this topic (page 13, line 429-434, and page 15, line 511-519).

  1. Warnakulasuriya, S.; Kerr, A. R., Oral Cancer Screening: Past, Present, and Future. J Dent Res 2021, 100, (12), 1313-1320.
  2. Chuang, S. L.; Su, W. W.; Chen, S. L.; Yen, A. M.; Wang, C. P.; Fann, J. C.; Chiu, S. Y.; Lee, Y. C.; Chiu, H. M.; Chang, D. C.; Jou, Y. Y.; Wu, C. Y.; Chen, H. H.; Chen, M. K.; Chiou, S. T., Population-based screening program for reducing oral cancer mortality in 2,334,299 Taiwanese cigarette smokers and/or betel quid chewers. Cancer 2017, 123, (9), 1597-1609.
  3. Cohen, N.; Fedewa, S.; Chen, A. Y., Epidemiology and Demographics of the Head and Neck Cancer Population. Oral Maxillofac Surg Clin North Am 2018, 30, (4), 381-395.
  4. Su, S. Y., Evaluation of Nationwide Oral Mucosal Screening Program for Oral Cancer Mortality among Men in Taiwan. Int J Environ Res Public Health 2022, 19, (21).
  5. Ministry of Health and Welfare Tobacco Hazards Prevention Act ,. https://law.moj.gov.tw/ENG/LawClass/LawAll.aspx?pcode=L0070021 (Mar 22),
  6. Ministry of Finance Tobacco and Alcohol Tax Act ,. https://law.moj.gov.tw/ENG/LawClass/LawAll.aspx?pcode=G0330010 (Mar 22),
  7. Tham, J.; Sem, G.; Sit, E.; Tai, M. C., The ethics of betel nut consumption in Taiwan. J Med Ethics 2017, 43, (11), 739-740.
  8. Yeh, C. Y.; Ho, L. M.; Lee, J. M.; Hwang, J. Y., The possible impact of an alcohol welfare surcharge on consumption of al-coholic beverages in Taiwan. BMC Public Health 2013, 13, 810.

Round 2

Reviewer 2 Report

Even though the authors revised the manuscript extensively, I still don't feel appropriate to accept it as they did not include the latest data rather, they just claim data were stable from 2019 to 2022. This shouldn't be a justification.